# Characterizing Massive Activations of Attention Mechanism in Graph Neural Networks

## Abstract

Graph Neural Networks (GNNs) have become increasingly popular for effectively modeling data with graph structures. Recently, attention mechanisms have been integrated into GNNs to improve their ability to capture complex patterns. This paper presents the first comprehensive study revealing a critical, unexplored consequence of this integration: the emergence of Massive Activations (MAs) within attention layers. We introduce a novel method for detecting and analyzing MAs, focusing on edge features in different graph transformer architectures. Our study assesses various GNN models using benchmark datasets, including ZINC, TOX21, and PROTEINS. Key contributions include (1) establishing the direct link between attention mechanisms and MAs generation in GNNs, (2) developing a robust definition and detection method for MAs based on activation ratio distributions, (3) introducing the Explicit Bias Term (EBT) as a potential countermeasure and exploring it as an adversarial framework to assess models robustness based on the presence or absence of MAs. Our findings highlight the prevalence and impact of attention-induced MAs across different architectures, such as Graph-Transformer, GraphiT, and SAN. The study reveals the complex interplay between attention mechanisms, model architecture, dataset characteristics, and MAs emergence, providing crucial insights for developing more robust and reliable graph models.

## 1 Introduction

Graph Neural Networks (GNNs) have emerged as a powerful tool for learning representations of graph-structured data, demonstrating remarkable success across various applications such as social network analysis (Min et al., 2021), recommendation systems (Gao et al., 2022) and molecular biology (Zhang et al., 2021). Central to the recent advancements in GNNs is the integration of attention mechanisms, which enable the models to focus on the most relevant parts of the input graph, thereby enhancing their ability to capture intricate patterns and dependencies.

Despite the substantial progress, the phenomenon of Massive Activations (MAs) within attention layers has not been thoroughly explored in the context of GNNs. MAs, characterized by exceedingly large activation values, can significantly impact the stability and performance of neural networks. In particular, understanding and mitigating MAs in GNNs is crucial for ensuring robust and reliable model behavior, especially when dealing with complex and large-scale graphs.

In this paper, we aim to bridge this gap by systematically investigating the occurrence and implications of MAs in attention-based GNNs. We focus on edge features in graph transformers, a state-of-the-art GNN architecture, and analyze how these features contribute to the emergence of MAs. Our study reveals that certain graph structures on edge configurations are more prone to inducing MAs, which in turn affects the overall performance and interpretability of the models.

To address these challenges, we propose a novel methodology for detecting and analyzing MAs in GNNs. Our approach involves a comprehensive evaluation of various GNN architectures, including GraphTransformer (Dwivedi & Bresson, 2021), GraphiT (Mialon et al., 2021), and SAN (Kreuzer et al., 2021), across multiple benchmark datasets, like ZINC (Irwin et al., 2012), TOX21 (Mayr et al., 2016; Huang et al., 2016) and OGBN-PROTEINS (Hu et al., 2020), which differs from their

downstream tasks like graph regression, multi-label graph classification, and multi-label node classification. We introduce specific criteria for identifying MAs and conduct extensive ablation studies to elucidate the role of edge features in this context.

This study represents the first comprehensive investigation of MAs in GNNs, laying the groundwork for future research. Our findings suggest that the scope of MAs analysis can be expanded to include a wider range of architectures and the evaluation of state-of-the-art attack methods, ultimately enhancing our understanding of MAs' influence on GNN performance and robustness. This is crucial for developing more robust and reliable graph transformer models, especially given the increasing popularity and widespread adoption of transformers in various applications today.

Our contributions are threefold:

- We provide the first systematic study on MAs in attention-based GNNs, highlighting their prevalence and impact on model performance.

- We propose a robust detection methodology for MAs, accompanied by detailed experimental protocols and ablation studies.

- We introduce the Explicit Bias Term (EBT) as a potential countermeasure for MAs, and we exploit it in an adversarial framework, called Explicit Bias Attack, to demonstrate the effectiveness of the MAs in compromising GNNs robustness.

Through this work, we aim to shed light on a critical yet understudied aspect of attention-based GNNs, offering valuable insights for the development of more resilient and interpretable graph-based models.

## 2 RELATED WORKS

GNNs have become effective instruments for studying and extracting insights from graph-structured data, with usages spanning fields like fraud detection (Motie & Raahemi, 2023), traffic prediction (Wang et al., 2022) and recommendation systems (Wu et al., 2021). The evolution of GNNs has been marked by significant advancements in their architectures and learning mechanisms, with a recent focus on incorporating attention mechanisms to enhance their expressive power and performance. The introduction of attention in GNNs was largely inspired by the success of transformers in natural language processing (Vaswani et al., 2017). Graph Attention Networks (GATs) (Veličković et al., 2017) were among the first to incorporate self-attention into GNNs, allowing nodes to attend differently to their neighbors based on learned attention weights. This innovation significantly improved the model's ability to capture complex relationships within graph structures.

Building upon the success of GATs, several variants and extensions have been proposed. GraphiT (Mialon et al., 2021) introduced a generalization of transformer architectures to graph-structured data, incorporating positional encodings and leveraging the power of multi-head attention mechanisms. Similarly, the Structure-Aware Network (SAN) (Kreuzer et al., 2021) proposed a novel attention mechanism that explicitly considers the structural properties of graphs, leading to improved performance on various graph-based tasks.

Recent studies on Large Language Models (LLMs) and Vision Transformers (ViTs) have revealed the presence of MAs within their internal states, specifically in the attention layer's output (Xiao et al., 2023; Sun et al., 2024). This phenomenon prompted investigations into the role of these activations in model behavior, performance, and potential vulnerabilities. Similar observations were made in Vision Transformers (ViTs) (Darcet et al., 2023; Dosovitskiy et al., 2020), suggesting that the presence of MAs might be a common feature in transformer-based architectures across different domains. These findings have led to a growing interest in understanding the implications of MAs for model interpretability, robustness, and potential vulnerabilities to adversarial attacks.

The study of internal representations in deep learning models has been a topic of significant interest in the machine learning community. Works such as Bau et al. (2020) have explored the interpretability of neural networks by analyzing activation patterns and their relationships to input features and model decisions. However, the specific phenomenon of MAs in GNNs has remained largely unexplored until now, representing a crucial gap in our understanding of these models.

The intersection of adversarial attacks and GNNs is another relevant area of study that relates to the investigation of MAs. Previous work has explored various attack strategies on graph data, including topology attacks, feature attacks, adversarial training and hybrid approaches (Sun et al., 2022a; Gosch et al., 2024). However, the potential vulnerabilities introduced by MAs represent a novel direction for research in this field. Understanding how MAs might be exploited or manipulated by adversarial inputs could lead to the development of more robust GNN architectures.

However, in the broader context of neural network analysis, techniques for probing and interpreting model internals have been developed. Methods such as feature visualization (Olah et al., 2017) and network dissection (Bau et al., 2017) have provided insights into the functions of individual neurons and layers in convolutional neural networks. Adapting and extending these techniques to analyze MAs in GNNs could provide valuable insights into their role and impact in possible future works.

Finally, the study of attention mechanisms in various neural network architectures has also yielded insights that may be relevant to understanding MAs in GNNs. Work on attention flow (Abnar & Zuidema, 2020) and attention head importance (Michel et al., 2019) in transformer models has shown that not all attention heads contribute equally to model performance, and some may even be pruned without significant loss of accuracy. These findings raise questions about whether similar patterns might exist in graph transformer models and how they might relate to the presence of MAs.

## 3 Terminology of Massive Activations in GNNs

Building upon the work on MAs in LLMs (Sun et al., 2024), we extend this investigation to GNNs, focusing specifically on graph transformer architectures. Our study encompasses various models, including GraphTransformer (GT) (Dwivedi & Bresson, 2021), GraphiT (Mialon et al., 2021), and Structure-Aware Network (SAN) (Kreuzer et al., 2021), applied to diverse task datasets such as ZINC, TOX21, and OGBN-PROTEINS (see Appendix A, B, C for details on models' configurations and datasets' composition). This comprehensive approach allows us to examine the generality of MAs across different attention-based GNN architectures.

### 3.1 Characterization of Massive Activations

MAs in GNNs refer to specific activation values that exhibit unusually high magnitudes compared to the typical activations within a layer. These activations are defined by the following criteria, where an activation value is intended to be its absolute value:

**Magnitude Threshold**: An activation is classified as massive if its value exceeds a predetermined threshold. This threshold is typically set to a value that is significantly higher than the average activation value within the layer, ensuring that only the most extreme activations are considered.

**Relative Threshold**: In the paper by Sun et al. (2024), MAs were defined as at least 1,000 times larger than the median activation value within the layer. This relative threshold criterion helped differentiate MAs from regular high activations that might occur due to normal variations in the data or model parameters.

The formal definition was represented as:

$$\text{MAs} = \{a \mid a > 100 \text{ and } a > 1000 \times \text{median}(\mathbf{A})\}$$

where $\mathbf{A}$ represents the set of activation values in a given layer.

However, in contrast to previous studies that employed a fixed relative threshold, our approach adopts a more rigorous method. We estimate MAs by comparing the distributions of activation ratios between a base, untrained model with Xavier weight initializations (Glorot & Bengio, 2010), and a fully trained model. This method ensures a more precise identification of MAs based on empirical data rather than an arbitrary fixed threshold. In this way, the untrained model serves as a reference for identifying unusual activations that emerge during training.

### 3.1.1 Detection Methodology

For both the base and trained models, we detected the MAs following a systematic procedure:

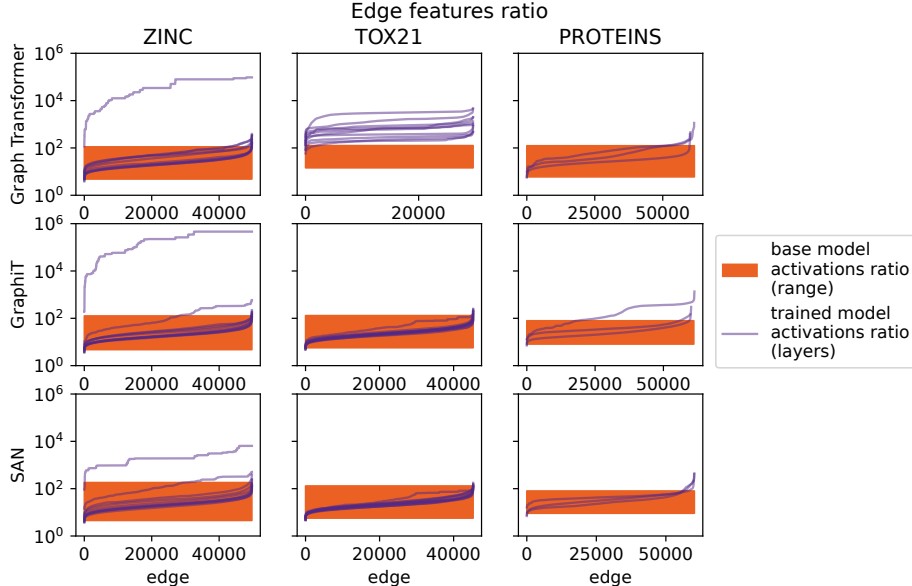

Figure 1: Comparison of MAs for trained vs base models, along all the edges. Activation values have been normalized within each layer by the layer's edge median. Represented ratios have been sorted increasingly for each layer independently.

**Normalization**: We normalized the activation values within each layer, dividing them by the edge median on the layer, to account for variations in scale between different layers and models. This normalization step ensures a consistent basis for comparison. The choice of dividing by the edge median comes from the huge amount of MAs being present, since almost every edge in the layers presenting MAs holds at least one MA, as shown from Figure 1. This is probably caused by the fact that attention is computed between pairs of adjacent nodes only, in contrast to LLMs where it is computed among each pair of tokens, therefore the model tends to spread MAs among almost all the edges to make them "available" to the whole graph. Indeed, Figure 1 indicates that MAs are a common phenomenon across different models and datasets, that they are not confined to specific layers but are distributed throughout the model architecture, and that MAs are an inherent characteristic of the attention-based mechanism in graph transformers and related architectures, not strictly dependent on the choice of the dataset.

**Batch Analysis**: We analyzed the activations on a batch-by-batch basis, minimizing the batch size, to have suitable isolation between the MAs and to ensure that the detection of MAs is not influenced by outliers in specific samples. For each activation, we computed the ratio of its magnitude to the edge median:

$$\text{ratio}(\text{activation}) = \frac{\text{abs}(\text{activation})}{\text{median}(\text{abs}(\text{edge\_activations}))} \tag{1}$$

and activations whose ratio exceeds the threshold are flagged as massive. Then, we considered the maximum ratio of each batch to detect those containing MAs.

**Layer-wise Aggregation**: We performed this analysis across multiple layers of the model to identify patterns and layers that are more prone to exhibiting MAs. This layer-wise aggregation helps in understanding the hierarchical nature of MAs within the model.

Figure 2 reports the analysis results. The batch ratios significantly increase in the trained transformers, concerning base ones, often even overcoming the threshold of 1000 defined by previous works (Sun et al., 2024), showing the presence of MAs in graph transformers, too.

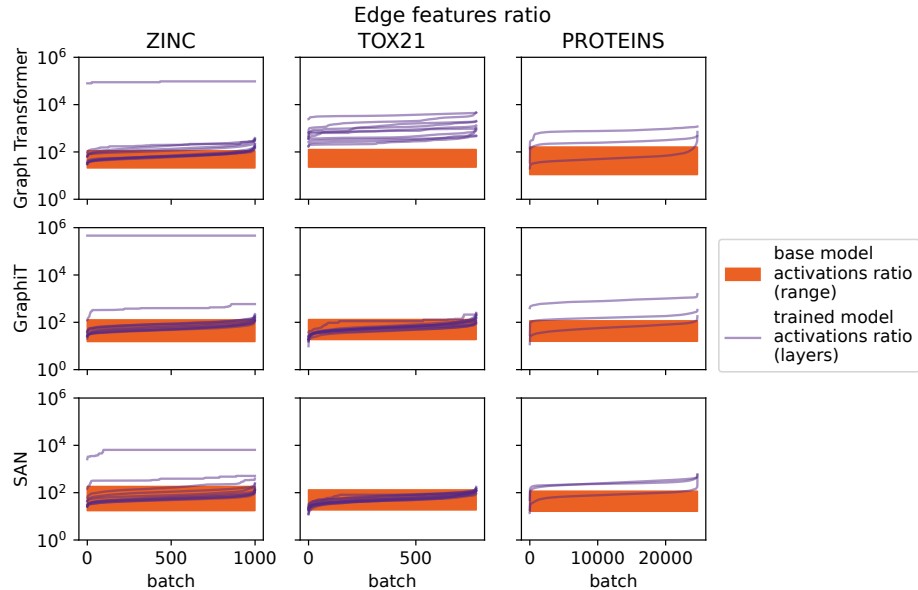

Figure 2: Comparison of MAs on trained against base models, without the use of Explicit Bias Term. Represented ratios have been sorted increasingly for each layer independently.

## 4 METHODOLOGY AND OBSERVATIONS

Focusing on edge features, first, we analyzed the ratio defined in Equation (1), taking the maximum for every batch, across the layers of each selected model and dataset, and visually compared the outcomes to value ranges obtained using the same model in a base state (with its parameters randomly initialized, without training) to verify the appearance of MAs. The graphical comparison, reported in Figure 2, shows ratios over the base range in most of the trained models, representing MAs.

To better characterize MAs, we studied their distribution employing the Kolmogorov-Smirnov statistic (Chakravarti et al., 1967). We found that a gamma distribution well approximates the negative logarithm of the activations' magnitudes, as well as their ratios. Figure 3a shows this approximation for a base model layer. We point out that, according to the existing definition, items on the left of the $-3$ are MAs.

We then compared the distributions of the $\log$-values between the base and trained models. Figure 3 illustrates this comparison, highlighting a significant shift in the distribution of the trained model compared to the base model. Moreover, this shift underscores the emergence of MAs during the training process, affirming that the threshold around $-\log(\text{ratio}) = -3$ (e.g., a ratio of 1000 or higher) effectively captures these significant activations, though sometimes it appears to be slightly shifted to the right as in Figure 3c.

When MAs appear, we have found two possible phenomenons:

- A lot of massive activation values are added on the left-hand side of the distribution, preventing a good approximation (Figure 3b).
- A few values appear on the left-hand side of the distribution, as spikes or humps or out-of-distribution values, which may or may not deteriorate the approximation, as shown in Figures 3c and 3d.

For example, histogram in Figure 3a represents the base model with untrained weights (only Xavier initialization). The gamma approximation fits the sample histogram well, with a low Kolmogorov-Smirnov (KS) statistic of $0.020$, indicating a very nice fit.

Figure 3b shows that the distribution of the trained model exhibits a significant shift due to a big hump appearing on the left side, representing extreme activation ratios (MAs). Indeed, the gamma

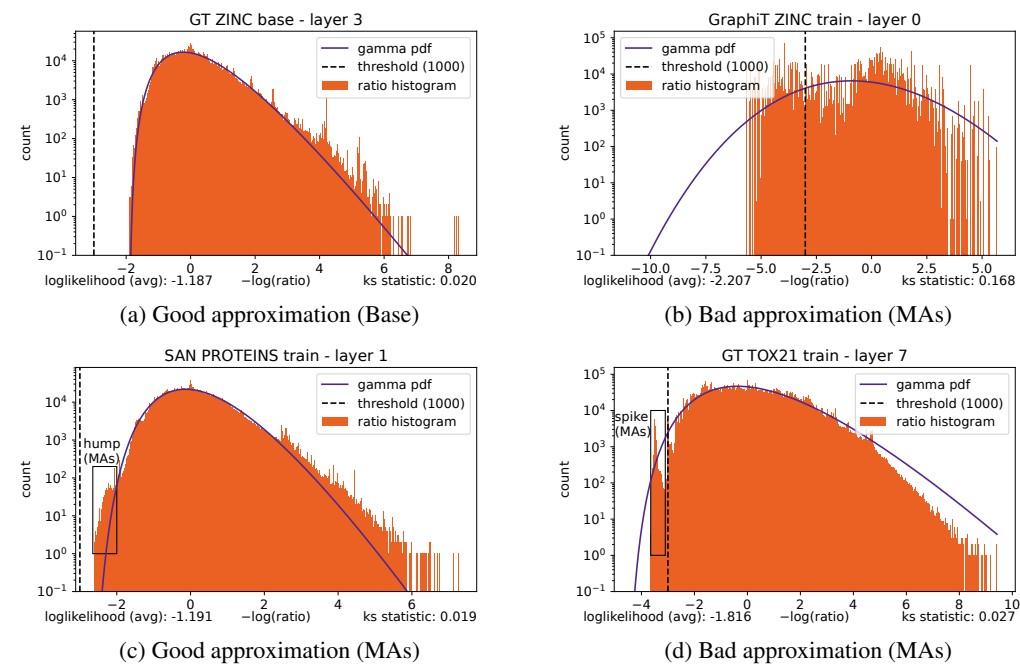

(a) Good approximation (Base)

(b) Bad approximation (MAs)

(c) Good approximation (MAs)

(d) Bad approximation (MAs)

Figure 3: Activation distributions for base and trained (with MAs) models. In Figure 3d we clearly distinguish a spike on the left of the distribution, corresponding to a ratio of 1000 (-log(ratio) = −3), which identifies the separation between the basic and massive regimes. The approximation pdf is rescaled to match the histogram scale.

approximation does not fit well, with a higher KS statistic of $0.168$, indicating a poor match caused by the presence of MAs.

Moreover, in the histogram of Figure 3d the trained model's distribution exhibits a clear spike on the left side at $-\log(\text{ratio}) = -3$, corresponding to a ratio of $1000$. This separation indicates the distinction between basic and massive activation regimes. The gamma distribution doesn't fit well this time, because of this spike preventing a good approximation, with a KS statistic of $0.027$ highlighting the model's shift due to training.

Figure 3c also shows the trained model's distribution, with a noticeable hump on the left side indicating MAs. The gamma approximation fits better than in Figure 3d, with a KS statistic of $0.019$, but still indicates the presence of MAs in the trained model, meaning that MAs have been added on the left-hand side of the distribution.

### 4.1 INSIGHTS AND IMPLICATIONS

From Figures 1 and 2 we can highlight the following points.

1. **Dataset Influence**:
   - The ZINC and OGBN-PROTEINS datasets consistently show higher activation values across all models compared to TOX21, suggesting that the nature of these datasets significantly influences the emergence of MAs. Even though many MAs are emerging form GT on TOX21.

2. **Model Architecture**:
   - Different GNN models exhibit varying levels of MAs. For instance, GraphTransformer and GraphiT tend to show more pronounced MAs than SAN, indicating that model architecture plays a crucial role.

3. **Impact of Attention Bias**:

Table 1: Comparison of test loss with and w/o bias for the different models and datasets. In bold the worst performances.

| Dataset | Model | Test loss | Test loss (EBT) |
|---|---|---|---|
| ZINC | GraphTransformer | 0.26 | **0.29** |
| | GraphiT | 0.13 | **0.31** |
| | SAN | 0.18 | **0.27** |
| TOX21 | GraphTransformer | 0.25 | **0.29** |
| | GraphiT | **0.38** | 0.32 |
| | SAN | **0.38** | 0.31 |
| OGBN-PROTEINS | GraphTransformer | **0.13** | 0.12 |
| | GraphiT | 0.14 | **0.16** |
| | SAN | 0.13 | 0.13 |

- Previous works suspect that MAs have the function of learned bias, showing that they disappear introducing bias at the attention layer. This holds for LLMs and ViTs, and for our GNNs as well, as shown in Figure 2 where the presence of MAs is affected by the introduction of the Explicit Bias Term on the attention. Figure 4 and text below suggest that MAs are intrinsic to the models' functioning, being anti-correlated with the learned bias.

The consistent observation of MAs in edge features, across various GNN models and datasets, points to a fundamental characteristic of how these models process relational information.

Inspired by recent advancements in addressing bias instability in LLMs (Sun et al., 2024), we introduced an Explicit Bias Term (EBT) into our graph transformer models. This bias term is discovered to counteract the emergence of MAs by stabilizing the activation magnitudes during the attention computation. The EBT is computed as follows:

$$\boldsymbol{b}_e = \boldsymbol{Q}\boldsymbol{k}\boldsymbol{e}' \tag{2}$$

$$\boldsymbol{b}_v = \mathrm{softmax}(\boldsymbol{A}_e)\boldsymbol{v}', \tag{3}$$

where $\boldsymbol{k}, \boldsymbol{e}, \boldsymbol{v} \in \mathbb{R}^d$ are the key, edge, and node bias terms (one per each attention head), $\boldsymbol{A}_e$ is the edge attention output, and $d$ the corresponding hidden dimension. $\boldsymbol{b}_e$ and $\boldsymbol{b}_v$ represent the edge and node bias terms and are added to the edge and node attention outputs, respectively. By incorporating EBT into the edge and node attention computations, and adding bias in the linear projections of the attention inputs, we regulated the distribution of activation values, thus mitigating the occurrence of MAs.

As shown in Figure 4, the introduction of these bias terms significantly reduces the frequency and magnitude of MAs, bringing the activation ratios closer to those observed in the base models. The effect of EBT is evident across all the different datasets. Whether it's ZINC, TOX21, or OGBN-PROTEINS, the activation ratios are brought closer to the baseline levels observed in the untrained models. This consistency underscores the general applicability of EBT in various contexts and downstream tasks. Moreover, Figure 4 shows that EBT mitigates MAs across different layers of the models. This is crucial as it indicates that EBT's effect is not limited to specific parts of the network but is extended throughout the entire architecture. For example, GraphTransformer on ZINC without EBT shows MAs frequently exceed $10^4$, while when EBT has been applied these ratios are significantly reduced, aligning more closely with the base model's range.

Table 1 shows that EBT does not systematically influence the test loss equally across different models and datasets. We have considered the test loss metric to keep the approach general, making it extendable to different downstream tasks. This ensures that the proposed method can be applied broadly across various applications of graph transformers.

Although the test loss remains relatively unchanged with the introduction of EBT, its presence helps in mitigating the occurrence of MAs, as evidenced by the reduction in extreme activation values

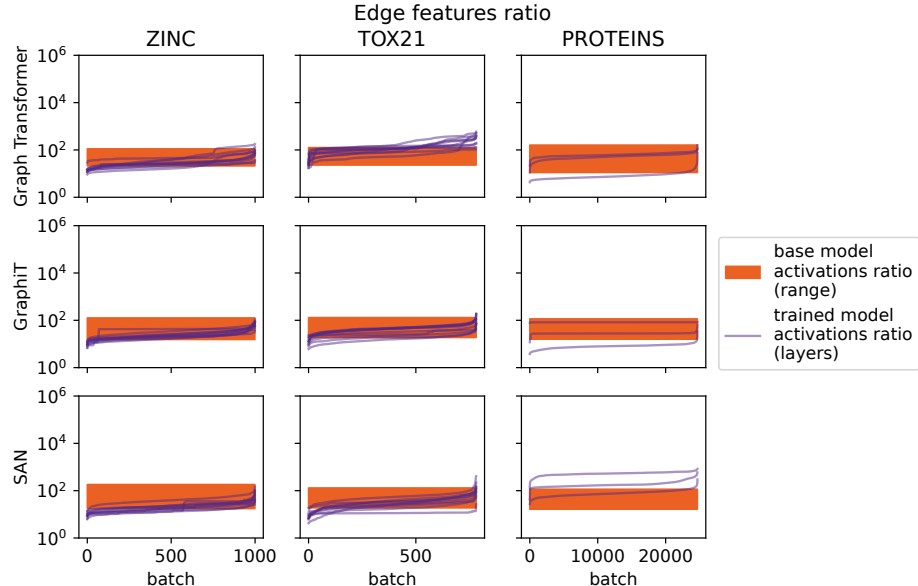

Figure 4: Comparison of MAs on trained against base models, with the use of Explicit Bias Term. Represented ratios have been sorted increasingly for each layer independently.

observed in earlier figures. By analyzing these results, it becomes evident that while EBT does not drastically alter the test performance, it plays a crucial role in controlling activation anomalies, thereby contributing to the robustness and reliability of graph transformer models.

In the next section, we will demonstrate how attacking the model with and without MAs can directly impact the robustness of the architectures. This will provide deeper insights into the robustness of graph transformers in the presence of MAs, suggesting their potential pitfalls.

## 5  EXPLICIT BIAS ATTACK

The study of adversarial attacks on GNNs has become increasingly important as these models are deployed in critical applications. While various attack strategies have been explored (Zügner et al., 2018; Sun et al., 2022b), the vulnerability introduced by MAs remains largely unexplored. Understanding how MAs can be exploited by adversaries is crucial for developing more robust GNN architectures and their downstream tasks. In this section, we propose the Explicit Bias Attack, a gradient-based method designed to exploit MAs and assess model robustness. Our approach is inspired by gradient ascent attacks previously applied to image classifiers (Goodfellow et al., 2014) and adapted for graph data (Dai et al., 2018). By analyzing the effectiveness of gradient ascent attack with and without the presence of EBT and MAs, we aim to provide insights into the role of these activations in model fragility.

Therefore, inspired by previous section, we exploited EBT as computed in Equations (2) and (3) to analyze the importance of MAs for a gradient ascent attack at test time, where noise (added to the input feature embedding) is learned to directly maximize the loss function. The effectiveness of an attack is evaluated by comparing the average test loss before and after the attack (i.e., with random and optimized noise, with the same standard deviations, respectively), using a gain defined as

$$\text{attack gain} = \frac{\text{optimized noise loss} - \text{random noise loss}}{\text{random noise loss}} \tag{4}$$

thus a higher gain means a more dangerous attack. We focus on GraphTransformer (GT) with TOX21 because the presence of MAs in each layer - as shown by Figure 2 - highlights the MAs effect for the attack, and compare the power of this method with and without the use of the EBT, which calls off the model's MAs.

Table 2: GT on TOX21 – Comparison of the noise optimization strategy with (no EBT) and without (EBT) MAs, due to the use of the explicit attention bias. The noise is optimized to maximize the loss function, and the results are shown for 1000 epochs of noise optimization.

| Noise dev. | Gain (no EBT, %) | Gain (EBT, %) |
|---|---|---|
| 0.01 | **1.50** | 1.41 |
| 0.03 | **1.83** | 1.78 |
| 0.10 | **4.73** | 2.53 |

Table 2 shows a stable increase of gain when dealing with MAs, using noise with standard deviation values of 0.01, 0.03, and 0.1 (the input feature embedding has a standard deviation of about 0.9) optimized for 1000 epochs on the test set. Table 2 highlights that MAs can be dangerous for the robustness of a model, and potentially exploited by attacks. These results indicate that a gradient ascent attack is effective in degrading model performance, especially in the presence of MAs. However, the introduction of explicit bias, consistent with the reduction of MAs, can significantly mitigate the impact of the attack, leading to more robust models. This highlights the importance of considering bias in designing defenses against these types of adversarial attacks, to prevent them from exploiting the presence of MAs.

In future work, to enable us to comprehensively assess the correlation between model robustness/fragility and the presence of MAs, we intend to delve deeper into different graph attack configurations while targeting MAs. This will offer a richer understanding of how these vulnerabilities can be mitigated, in favor of more reliable models.

# 6 CONCLUSION AND FUTURE WORK

This paper presents the first comprehensive study of MAs in attention-based GNNs. We have introduced a novel methodology for detecting and analyzing MAs, focusing on edge features in various graph transformer architectures across multiple benchmark datasets. Our findings reveal that MAs are prevalent across different models and datasets, and demonstrate that they could be effectively leveraged by adversaries to degrade the performance of GNNs.

We showed that the introduction of Explicit Bias Terms (EBT) can effectively mitigate the occurrence of MAs, leading to more stable activation distributions. However, our results also showed that this mitigation does not always translate to improved test performance, highlighting the complex role of MAs in GNNs' behavior.

Furthermore, we introduced the Explicit Bias Attack, a gradient-ascent adversarial framework, that demonstrates how MAs, if not mitigated by EBT, can expose models to vulnerabilities in their tasks. This further points out the importance of considering these activations in the context of model robustness.

Future research will expand this analysis to a wider range of architectures and advanced attack methods, further clarifying the influence of MAs on GNN performance and robustness, and potentially leading to more interpretable and stable graph-based models. Specifically, future research could explore:

- **Customized Adversarial MAs**: Developing more adversarial techniques to regulate and attack these activations to enhance model stability and performance, like injecting fake MAs or exploiting state-of-the-art graph attack methods.

- **Downstream-driven MAs**: Leveraging MAs for specific downstream task, investigating how to harness these significant activations to improve models and their interpretability on specific assignments such as link prediction or drug design.

- **Comparative Analysis**: Extending the study to additional models and datasets to generalize the findings further and uncover broader patterns.

These insights provide a deeper understanding of the internal mechanisms of attention-based GNNs and highlight the way for improvements in graph learning models. By addressing the challenges and opportunities presented by MAs, we can work towards developing more robust, interpretable, and effective GNN architectures for a wide range of applications.

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

# A   Dataset Composition

This section provides additional details on the used datasets throughout the experiments.

The ZINC dataset (Irwin et al., 2012) is a benchmark collection for evaluating GNNs in molecular chemistry, where molecules are represented as graphs with atoms as nodes and chemical bonds as edges. Contents include:

- Graphs: The dataset includes over $250,000$ molecular graphs. Each molecule is represented by a graph with nodes (atoms) and edges (bonds), incorporating various bond types (e.g., single, double, triple).

- Node Features: Atoms are described by features that capture their chemical properties, such as atom types, hybridization states, and other atomic attributes.

- Edge Features: Bonds between atoms are characterized by features representing bond types and additional chemical information.

- Task: The primary task is **graph regression**, where the goal is to predict continuous values associated with each molecule. This often involves predicting molecular properties such as solubility or biological activity.

ZINC is useful for evaluating GNNs' performance in learning molecular representations and predicting continuous chemical properties, providing insights into the model's ability to generalize across diverse chemical compounds.

The TOX21 dataset (Mayr et al., 2016; Huang et al., 2016) is designed for toxicity prediction and focuses on classifying chemical compounds based on their potential toxicity. It is part of the Toxicology Data Challenge and features molecular graphs with associated toxicity labels. Contents include:

- Graphs: The dataset consists of molecular graphs where nodes represent atoms and edges represent chemical bonds. It includes thousands of molecules with toxicity annotations, and it consists of $7,831$ graphs with each graph representing a molecular structure with associated toxicity labels.

- Node Features: Atoms are encoded with features representing their types, hybridization states, and other chemical properties.

- Edge Features: Bonds are detailed with features indicating bond types and additional chemical attributes.

- Task: The main task is **multi-label graph classification**, where each molecule is classified into multiple toxicity categories. This allows for the prediction of various toxicity endpoints simultaneously.

TOX21 is valuable for assessing GNN models in predicting toxicity from molecular structures, which is crucial for drug discovery and safety evaluation, providing a benchmark for multi-label classification tasks.

The OGBN-PROTEINS dataset, part of the Open Graph Benchmark (OGB) (Hu et al., 2020), focuses on protein function prediction. It contains one large graph representing protein structures, with nodes corresponding to amino acids and edges to their interactions. Contents include:

- One Large Graph: OGBN-PROTEINS contains $54,879$ nodes and $89,724$ edges. These nodes represent amino acids in protein structures, and edges represent interactions or bonds between these amino acids. It includes various protein structures used for functional prediction.

- Node Features: Amino acids are described by features capturing biochemical properties, such as amino acid type, secondary structure, and other relevant attributes.

- Edge Features: Edges denote interactions between amino acids and include features reflecting the nature of these interactions or spatial relationships.

- Task: The task is **multi-label node classification**, where the goal is to predict multiple functional categories for each amino acid node in the protein graph. This involves classifying nodes into various functional classes based on their role in the protein's functionality.

OGBN-PROTEINS is suitable for evaluating GNNs on biological data, specifically in predicting protein functions based on structural information. It provides insights into how well models can handle multi-label node classification tasks in a complex biological context.

## B  MODEL ARCHITECTURE

This section provides additional details on the models' architecture used throughout all the experiments, namely GT (Dwivedi & Bresson, 2021), GraphiT (Mialon et al., 2021) and SAN (Kreuzer et al., 2021). These graph-transformer architectures integrate the principles of both GNNs and transformers, leveraging the strengths of attention mechanisms to capture intricate relationships within graph-structured data. Graph transformers extend the transformer structure, typically used for sequence data, to graphs, operating by embedding nodes and edges into higher-dimensional spaces and then applying multi-head self-attention mechanisms to capture dependencies between nodes.

Mathematically, let $\mathcal{G} = (V, E)$ be a graph where $V = \{v_1, ..., v_n\}$ is the set of nodes and $E \subseteq V \times V$ is the set of edges. Each node $v_i$ is associated with a feature vector $\boldsymbol{x}_i \in \mathbb{R}^d$, and each edge $(v_i, v_j)$ may have an edge feature $\boldsymbol{e}_{ij} \in \mathbb{R}^k$. Therefore, graph transformer models are designed as follows.

### INPUT EMBEDDING

The initial node features $\boldsymbol{X} = [\boldsymbol{x}_1, ..., \boldsymbol{x}_n]^T \in \mathbb{R}^{n \times d}$ are typically projected to a higher-dimensional space:

$$\boldsymbol{H}^{(0)} = \boldsymbol{X}\boldsymbol{W}_{in} + \boldsymbol{b}_{in} \tag{5}$$

where $\boldsymbol{W}_{in} \in \mathbb{R}^{d \times d'}$ is a learnable weight matrix and $\boldsymbol{b}_{in} \in \mathbb{R}^{d'}$ is a bias vector.

### POSITIONAL ENCODING

To capture structural information, positional encodings $\boldsymbol{P} \in \mathbb{R}^{n \times d'}$ are often added:

$$\boldsymbol{H}^{(0)} = \boldsymbol{H}^{(0)} + \boldsymbol{P} \tag{6}$$

### MULTI-HEAD ATTENTION LAYER

The core of a graph transformer is the multi-head attention mechanism. For each attention head $i$ (out of $h$ heads) there are also:

1. Query, Key, and Value Projections:

$$\boldsymbol{Q}_i = \boldsymbol{H}^{(l)}\boldsymbol{W}_i^Q \tag{7}$$
$$\boldsymbol{K}_i = \boldsymbol{H}^{(l)}\boldsymbol{W}_i^K \tag{8}$$
$$\boldsymbol{V}_i = \boldsymbol{H}^{(l)}\boldsymbol{W}_i^V \tag{9}$$

where $\boldsymbol{W}_i^Q, \boldsymbol{W}_i^K, \boldsymbol{W}_i^V \in \mathbb{R}^{d' \times d_k}$ are learnable weight matrices, and $d_k = d'/h$.

2. Attentions Scores (node features only):

$$\boldsymbol{A}_i = \text{softmax}\left(\frac{\boldsymbol{Q}_i\boldsymbol{K}_i^T}{\sqrt{d_k}} + \boldsymbol{M}\right), \tag{10}$$

where $\boldsymbol{M} \in \mathbb{R}^{n \times n}$ is a mask matrix to enforce the graph structure:

$$M_{i,j} = \begin{cases} 0 & \text{if } (v_i, v_j) \in E \text{ or } i = j \\ -\infty & \text{otherwise.} \end{cases} \tag{11}$$

3. Output of each head:

$$\textbf{head}_i = \boldsymbol{A}_i\boldsymbol{V}_i. \tag{12}$$

4. Concatenation and Projection:

$$\boldsymbol{H}' = \text{Concat}(\textbf{head}_1, ..., \textbf{head}_h)\boldsymbol{W}^O, \tag{13}$$

where $\boldsymbol{W}^O \in \mathbb{R}^{d' \times d'}$ is a learnable weight matrix.

### FEED-FORWARD NETWORK (FFN)

Each attention layer is typically followed by a position-wise feed-forward network:

$$\text{FFN}(\boldsymbol{x}) = \max(0, \boldsymbol{x}\boldsymbol{W}_1 + \boldsymbol{b}_1)\boldsymbol{W}_2 + \boldsymbol{b}_2 \tag{14}$$

where $\boldsymbol{W}_1 \in \mathbb{R}^{d' \times d_{ff}}$, $\boldsymbol{W}_2 \in \mathbb{R}^{d_{ff} \times d'}$, $\boldsymbol{b}_1 \in \mathbb{R}^{d_{ff}}$, and $\boldsymbol{b}_2 \in \mathbb{R}^{d'}$ are learnable parameters.

### LAYER NORMALIZATION AND RESIDUAL CONNECTIONS

Each sub-layer (attention and FFN) employs a residual connection followed by layer normalization:

$$\boldsymbol{H}^{(l+1)} = \text{LayerNorm}(\boldsymbol{H}^{(l)} + \text{Sublayer}(\boldsymbol{H}^{(l)})) \tag{15}$$

where Sublayer is either the multi-head attention or the FFN.

### EDGE FEATURE INTEGRATION

GraphTransformer, GraphiT and SAN incorporate edge features:

1. In attention computation:

$$A_{i,j} = \text{softmax}\left(\frac{\boldsymbol{q}_i^T \boldsymbol{k}_j + f(\boldsymbol{e}_{ij})}{\sqrt{d_k}}\right) \tag{16}$$

where $f$ is a learnable function (e.g., a small neural network) that projects edge features.

2. In value computation:

$$\boldsymbol{v}_{ij} = \boldsymbol{V}_i + g(\boldsymbol{e}_{ij}) \tag{17}$$

where $g$ is another learnable function.

### GLOBAL NODE

Some architectures introduce a global node $v_g$ connected to all other nodes to capture graph-level information:

$$\boldsymbol{h}_g^{(l+1)} = \text{Attention}(\boldsymbol{h}_g^{(l)}, \boldsymbol{H}^{(l)}) \tag{18}$$

### OUTPUT LAYER

The final layer depends on the task:

- For node classification: $\boldsymbol{y}_{node} = \text{softmax}(\boldsymbol{H}_{node}^{(L)}\boldsymbol{W}_{out} + \boldsymbol{b}_{out})$
- For graph classification: $\boldsymbol{Y}_{graph} = \text{MLP}(\text{Pool}(\boldsymbol{H}^{(L)}))$

where Pool is a pooling operation (e.g., mean, sum, or attention-based pooling) to switch from single node to graph embedding level.

### TRAINING

The model is typically trained end-to-end using backpropagation to minimize a task-specific loss function, such as cross-entropy for classification or mean squared error for regression.

## C   KOLMOGOROV-SMIRNOV TEST

This section provides additional details on the Kolmogorv-Smirnov (KS) test (Chakravarti et al., 1967) used to analyze the distribution of activations. The KS test is a non-parametric test that compares the cumulative distribution functions of two samples. It is used to compare a sample with a reference probability distribution (one-sample KS test) or to compare two samples (two-sample KS test) with each other.

In our study, we utilized the KS statistic to compare the distribution of activation values before and after training (i.e. base against trained model), identifying Massive Activations (MAs). We primarily used the one-sample KS test to assess the goodness of fit between our observed activation distributions and a theoretical gamma distribution.

## C.1 ONE-SAMPLE KOLMOGOROV-SMIRNOV TEST

The one-sample KS test can typically be formulated as follows:

### C.1.1 NULL HYPOTHESIS

The null hypothesis for the one-sample KS test is:

$H_0$: The sample data follows the specified distribution (in our case, a gamma distribution).

### C.1.2 TEST STATISTIC

The KS statistic $D_n$ is defined as the supremum of the absolute difference between the empirical cumulative distribution function (ECDF) $F_n(x)$ of the sample and the cumulative distribution function (CDF) $F(x)$ of the reference distribution:

$$D_n = \sup_x |F_n(x) - F(x)| \tag{19}$$

where $\sup_x$ denotes the supremum of the set of distances.

### C.1.3 EMPIRICAL CUMULATIVE DISTRIBUTION FUNCTION

For a given sample $x_1, x_2, ..., x_n$, the ECDF is defined as:

$$F_n(x) = \frac{1}{n} \sum_{i=1}^{n} \mathbf{1}_{x_i \leq x} \tag{20}$$

where $\mathbf{1}_{x_i \leq x}$ is the indicator function, equal to 1 if $x_i \leq x$ and 0 otherwise.

### C.1.4 CRITICAL VALUES AND P-VALUE

The distribution of the KS test statistic under the null hypothesis can be calculated, which allows us to obtain critical values and p-values. The null hypothesis is rejected if the test statistic $D_n$ is greater than the critical value at a chosen significance level $\alpha$, or equivalently if the p-value is less than $\alpha$.

## C.2 APPLICATION TO MAs DETECTION

In our experiments, we used the KS statistic to assess whether the distribution of activation ratios in our GNNs follows a gamma distribution. The process is as follows:

1. We computed the activation ratios for each layer of our models, as defined in Equation (1) of the main paper.

2. We took the negative logarithm of these ratios to transform the distribution.

3. We fit a gamma distribution to this transformed data using maximum likelihood estimation.

4. We performed a one-sample KS test to compare our sample data to the fitted gamma distribution.

The KS test statistic provides a measure of the discrepancy between the observed distribution of activation ratios and the theoretical gamma distribution. A lower KS statistic indicates a better fit, suggesting that the activation ratios more closely follow the expected distribution.

## C.3 INTERPRETATION IN THE CONTEXT OF MAs

Following the described procedure in Section C.2, we employed the KS statistic as quantitative/statistical measure to detect the presence of MAs:

- For untrained (base) models, we typically observed low KS statistics, indicating that the activation ratios closely follow a gamma distribution.

- For trained models exhibiting MAs, we often saw higher KS statistics. This indicates a departure from the gamma distribution, which we interpret as evidence of MAs.

- The magnitude of the KS statistic provided a quantitative measure of how significantly the presence of MAs distorts the expected distribution of activation ratios.

Moreover, we complemented our KS statistic results with visual inspections of the distributions and other analyses as described in the main paper.

