# OpenReview forum: "Characterizing Massive Activations of Attention Mechanism in Graph Neural Networks"
_ICLR.cc/2025/Conference — Submitted to ICLR 2025_

### Official Review · Reviewer_o79Q · 2024-11-01

**Soundness:** 2
**Presentation:** 2
**Contribution:** 2
**Rating:** 3
**Confidence:** 3

**Summary:**

This paper investigates the emergence of Massive Activations (MAs) in attention-based Graph Neural Networks (GNNs). It identifies the link between attention mechanisms and MAs, which can destabilize models. The study proposes a method to detect and analyze MAs, introduces the Explicit Bias Term (EBT) as a countermeasure, and evaluates various GNN architectures like GraphTransformer and GraphiT on benchmark datasets. The findings emphasize the impact of MAs on model performance and robustness, offering insights for developing more resilient graph-based models.

**Strengths:**

1.	The paper provides a comprehensive study on the emergence of Massive Activations (MAs) in attention-based GNNs, highlighting a previously unexplored issue that affects model stability and performance.
2.	The writing is logically clear and coherent.

**Weaknesses:**

1.	The contribution is overstated. The core observation of Massive Activations (MAs) and the proposed solution, Explicit Bias Term (EBT), are not new. The work primarily transfers concepts from prior work [1] on LLMs to attention-based GNNs.
2.	The observations of MAs on attention-based GNNs lack deeper analysis. For instance, while massive activations are suggested to serve as an important bias in the attention mechanism, the paper mainly discusses their focus on unimportant features without thorough examination.
3.	The technical motivation is not clear. The authors do not clearly explain the fundamental advantage between using an untrained model to estimate MAs and using the median activation value within the layer for estimation.
4.	All the figures and tables in the paper are not up to standard. The figures in the paper lack detailed explanations of the axes and the different lines, which may confuse readers.
5.	The experimental results are not convincing, as the tables do not show the variance across multiple trials.

**Questions:**

1.	What is the main contribution of this paper compared to the previous work [1]? It’s important to identify what new insights or advancements this paper offers.
2.	On page 5, line 258, the authors mention, "When MAs appear, we have found two possible phenomena." This statement, “possible phenomena,” may leave readers confused. The paper could benefit from a more detailed mechanistic explanation of these observations.
3.	On page 4, line 197, the authors state, "MAs are an inherent characteristic of the attention-based mechanism in graph transformers and related architectures, not strictly dependent on the choice of the dataset." However, observations from Figure 1 suggest that the same architecture exhibits significant variations in MAs across different datasets. This raises questions about the potential dependency of MAs on the dataset, which should be addressed more thoroughly.

[1] Sun M, Chen X, Kolter J Z, et al. Massive activations in large language models[J]. arXiv preprint arXiv:2402.17762, 2024.

---

> ### Author Response · Authors · 2024-11-22
>
> The authors thank Reviewer o79Q for all the interesting questions.
>
> ### Weakness/Question #1
> Although  [[1]](https://arxiv.org/abs/2402.17762) explored MAs for LLMs, our work differs in:
> - being the first work to study them on attention-based GNN;
> - employing statistical tools, such as KS test and statistics, and gamma distribution, to characterize them;
> - treating them as anomalies with respect to the base distribution;
> - defining the base model as the untrained model with Xavier randomly initialized weights;
> - overviewing MAs implication on models' robustness (Eq.4 - Table 2).
>
> ### Weakness #2
> We thank the reviewer for raising these points, however, could you please point out what "unimportant features" are?
>
> ### Weakness #3
> We are committed to integrating into the Introduction section a deeper explanation of the methodology of spotting MAs. We specify that we used the base (untrained) model to define what a standard behavior is for activation values, and that we used the median activation value to normalize the activation values in a given layer. Therefore, the median normalization is done for layers of both the trained and base models, whose distributions are then compared between each other.
>
> ### Weakness #4 - #5
> We appreciated the suggestion and we will modify tables and illustrations to meet the standards.
>
> ### Question #2
> Thank you again for raising this point. To provide an adequate response, we would like to know, according to the reviewer's opinion, what confusion the statement could generate.
>
> ### Question #3
> Indeed, the behavior of MAs changes depending on the choice of dataset, however, the values of activations in the trained model generally exceed the range of the base model (without MAs).
>
> ### References
> [1] Mingjie Sun, Xinlei Chen, J Zico Kolter, and Zhuang Liu. Massive activa-
> tions in large language models. arXiv preprint arXiv:2402.17762, 2024.

---

> > ### Author Response · Authors · 2024-11-26
> >
> > Thank you for taking the time to provide your feedback. We wanted to kindly check if you have had the opportunity to review our rebuttal and if there are any remaining concerns that we may not have addressed.

---

> > > ### Author Response · Authors · 2024-11-28
> > >
> > > Thank you for taking the time to provide your feedback. As we approach the deadline, we wanted to check if you have had the opportunity to review our rebuttal and if there are any remaining concerns we may not have addressed.

---

### Official Review · Reviewer_u2zz · 2024-11-03

**Soundness:** 3
**Presentation:** 2
**Contribution:** 2
**Rating:** 5
**Confidence:** 4

**Summary:**

Graph neural networks (GNNs) are popular for effectively modeling data with graph structures. Recently, attention mechanisms have been integrated into GNNs to improve their ability to capture complex patterns. This paper presents a comprehensive study on the emergence of massive activations (MAs) within attention layers. This study assesses various GNN models on three datasets, ZINC, TOX21, and PROTEINS.

**Strengths:**

1. This work studies the existence of massive activations (MAs) in attention layers of graph neural networks, improving the understanding of attention in graph structures.
2. This work presents various types of visualization, providing a deep insight on the existence of MAs and their distributions.
3. The observations and insights presented in this work can be beneficial to the community, considering that many people are working on improving a graph transformer structure.

**Weaknesses:**

1. In such a benchmark study, I would expect a lot more combinations of models, datasets, and configurations to verify the generalizability of observations and insights. I don’t believe the current experiments are comprehensive enough to have conclusive claims. It would be also nice to include more recent graph transformer structures.
2. This work is not motivated well, as it is not clear why we should study MAs, and why they are problematic. How exactly do they affect the final performance on downstream tasks? The authors provide Tables 1 and 2 in this regard, but they seem to be indirect evidence.
3. The authors claim that they propose a robust detection methodology and the explicit bias term as a solution for MAs, but I’m not certain if any of these make a significant contribution. Overall, I believe the technical quality of the paper should be improved, and the insights presented in the paper should be connected to what happens in the actual models.

**Questions:**

1. In lines 191 - 194, the authors mention that “attention is computed between pairs of adjacent nodes only,” but many graph transformers compute attention also for non-adjacent node pairs. Do the authors consider only the model structures with this constraint?

---

> ### Author Response · Authors · 2024-11-22
>
> The authors thank Reviewer u2zz for all the interesting questions.
> ### Weakness #1
> Due to similar questions, we report hereafter the same answer as Reviewer 5kwJ.
>
> We experimented with newer models such as GraphGPS [[1]](https://arxiv.org/abs/2205.12454) and CSA [[2]](https://arxiv.org/abs/2304.10933), which showed us the presence of MAs on both ZINC (as depicted in Figure 3 in `supplementary_material/ICLR25_Rebuttal.pdf`) and PCQM4Q-full v2 datasets. In addition, we will enrich the benchmark by introducing new models and datasets to further show the presence of MAs, e.g we will investigate [[1]](https://arxiv.org/abs/2205.12454) and [[2]](https://arxiv.org/abs/2304.10933) over:
> - ogbg-molhiv
> - ogbg-molpcba
> - ogbg-ppa
> - ogbg-code2
>
> The choice of models and datasets, in any way, always reflects our original idea: that is, models that incorporate extra information about edges from their datasets, via edge features, and thus focus attention on both nodes and edges during their message passing.
>
> ### Weaknesses #2 - #3
> As suggested in "Conclusion and Future Work", we acknowledge the importance of exploring the impact of MAs on various downstream tasks (e.g., customized adversarial MAs, downstream-driven MAs). However, being the first work to study them on graphs we have chosen to explore deeper the characterization of MAs and whether their presence can be spotted using statistical tests (KS-test have been used indeed) and rigorous empirical analysis. In relation to the latter, we are committed to investigate more datasets and configurations as stated in point #1.
>
> ### Question #1
> Indeed, the proposed models compute attention between pairs of adjacent nodes only. However, recent experiments with newer models such as GraphGPS [[1]](https://arxiv.org/abs/2205.12454) and CSA [[2]](https://arxiv.org/abs/2304.10933) (see Figure 3 in `supplementary_material/ICLR25_Rebuttal.pdf`), which show the presence of MAs on ZINC and PCQM4Q-full-v2, compute attention also for non-adjacent node pairs.
>
> ### References
> [1] Ladislav Ramp´aˇsek, Michael Galkin, Vijay Prakash Dwivedi, Anh Tuan Luu,
> Guy Wolf, and Dominique Beaini. Recipe for a general, powerful, scalable
> graph transformer. Advances in Neural Information Processing Systems,
> 35:14501–14515, 2022.
>
> [2] Romain Menegaux, Emmanuel Jehanno, Margot Selosse, and Julien Mairal.
> Self-attention in colors: Another take on encoding graph structure in trans-
> formers. arXiv preprint arXiv:2304.10933, 2023

---

> > ### Comment · Reviewer_u2zz · 2024-11-23
> >
> > I appreciate your response and the new experiments, but I’m not sure they are sufficient to change my opinion. I believe the work requires significant revisions and improvements to enhance its technical quality.

---

### Official Review · Reviewer_Hp9k · 2024-11-04

**Soundness:** 2
**Presentation:** 3
**Contribution:** 3
**Rating:** 5
**Confidence:** 4

**Summary:**

This paper investigates an interesting and significant phenomenon in attention-based Graph neural networks (GNNs): Massive activations (MAs) in attention layers. The authors conduct a systematic analysis of this phenomenon in GNNs built with different Graph Transformer layers and propose a method for detecting MAs and analyzing their effects on edge features. Besides, the Explicit Bias Term (EBT) is proposed as a countermeasure, whose robustness bringing to attention-based GNNs is also validated on well-established datasets.

**Strengths:**

1. The phenomenon investigated in this paper is significant, and the exploration of it may influence the performance of attention-based GNNs.
2. Some empirical experiments are comprehensive.

**Weaknesses:**

1. The motivations of this paper are not clear enough. The authors claim that their focus is variants of Graph Transformers. Why conventional attention-based GNNs (e.g., GATs and corresponding variants) are not considered in this work is not mentioned in the Section of Introduction.
2. Some terms are not well defined or explained. For example, what does the edge feature stand for in this paper?
3. The insights and observations presented in this paper are mainly based on the analysis of three test datasets, i.e., ZINC, TOX21, and PROTEINS. I’m wondering whether similar observations or insights can be obtained from a wide range of test datasets. The authors are suggested to conduct a more comprehensive analysis on more real-world datasets, e.g., ones from OGBN (for graph property prediction) and other publicly available datasets for graph-level tasks.
4. More recent attention-based GNNs should be investigated in the paper, which may provide an overview of the advances of graph attention. For example, GATv2, Graph conjoint attentions networks, and Adaptive structural fingerprint GNN should be investigated in the manuscript.
5. Some experiments can be extended. For example, the ablation studies shown in Table 2 can be extended to all three variants of Graph Transformers on all test datasets considered in this paper.

**Questions:**

1. Why are conventional attention-based GNNs not considered in this work?
2. What does the edge feature stand for in this paper?
3. Are similar observations or insights obtained from a wide range of test datasets?
4. More recent attention-based GNNs should be investigated in the paper.
5. Some experiments, e.g., the ablation studies in Table 2 can be extended.

**Details Of Ethics Concerns:**

This reviewer has no critical ethical concerns.

---

> ### Author Response · Authors · 2024-11-22
>
> The authors thank Reviewer Hp9k for all the interesting questions.
>
> ### Weakness/Question #1
> The authors are committed to further specify in the Introduction Section that the main motivation of the article lies in the identification of MAs in graph transformers models capable of using dataset's edge features in their attention mechanisms, so that additional information is considered in the message-passing. Therefore, conventional attention-based GNNs (e.g., GATs and corresponding variants) lacking edge features attention are not analyzed.
>
> ### Weakness/Question #2
> Edge features in these contexts refer to additional attributes or properties associated with the edges of the graphs. They represent information about the relationship or interaction between the connected nodes. Typically, edge features are specific to a domain, e.g., bond types in molecular graphs like ZINC and TOX21, or spatial/interaction properties between amino acids or secondary structure elements like PROTEINS.
>
> ### Weakness/Question #3
> Further experiments will be conducted extensively on other models and datasets, for example, we attach the results obtained for CSA-GraphGPS on PCQM4Mv2-full (from OGB) and ZINC (see Figure 3 in `supplementary_material/ICLR25_Rebuttal.pdf`). In addition, we will also consider other OGBN datasets, as already done in the paper with PROTEINS (which belongs precisely to OGBN for node prediction properties [OGBN-PROTEINS](https://ogb.stanford.edu/docs/nodeprop/#ogbn-proteins)).
>
> ### Weakness/Question #4
> Accordingly to the answer #1 and #3 we attach results of more recent edge-features attention-based graph transformers on PCQM4Mv2-full and ZINC datasets.
>
> ### Weakness/Question #5
> To have insights on the correlation between MAs and models' robustness, adversarial attack was carried out only on the most susceptible model-dataset configuration which is the one that exhibits MAs within each layer. Furthermore, authors want to specify that adversarial attack is proposed as possible future directions to be analyzed, and not meant to be the main focus at this stage, but rather to be deeper studied in follow-up works.

---

> > ### Comment · Reviewer_Hp9k · 2024-11-25
> >
> > Dear Authors, thanks very much for your prompt reply to my paper review. However, some responses are not convincing enough for me to change the scoring of this paper. For example, conventional graph attention networks can also incorporate edge features in some applications. Thus, the authors are suggested to explicitly discuss and enhance the motivations of the paper. Besides, most of the suggested experiments to test the effectiveness of the proposed approach are not conducted and presented during the rebuttal period.

---

### Official Review · Reviewer_5kwJ · 2024-11-04

**Soundness:** 2
**Presentation:** 2
**Contribution:** 2
**Rating:** 3
**Confidence:** 3

**Summary:**

This paper identifies the existence of Massive Activations (MAs) across various graph transformer models, emphasizing their adverse effects on model stability and performance. The paper proposes a novel mechanism called the Explicit Bias Term (EBT) that compares activation ratios between untrained and trained models to detect MAs. Subsequently, EBT is designed to stabilize activation values by integrating bias terms directly into the attention mechanism. Additionally, the study investigates the role of MAs in adversarial attacks, introducing a gradient ascent attack method to rigorously evaluate EBT’s effectiveness in mitigating MAs. Experimental results demonstrate that MAs are widely present across different GNN architectures and datasets, while EBT significantly reduces the impact of MAs and enhances model robustness.

**Strengths:**

1.	The paper employs a rigorous approach to define Massive Activations (MAs) and experimentally confirms that MAs are a widespread phenomenon across various attention-based models and datasets.
2.	The paper proposes the Explicit Bias Term (EBT) to mitigate the influence of Massive Activations (MAs), and this method is applicable across various downstream tasks.
3.	Experiments have shown that EBT can significantly reduce the influence of MAs and has proven to be effective in enhancing model robustness against the gradient ascent attack method.

**Weaknesses:**

1.	The paper introduces the Explicit Bias Term (EBT) as a countermeasure to reduce the impact of Massive Activations (MAs). However, it lacks sufficient theoretical justification or a detailed explanation of how EBT can effectively mitigate MAs.
2.	The paper presents the fitting results of the logarithm of activation distributions in Figure 3, but it uses the same gamma distribution for different datasets, models (GraphTransformer, GraphiT, SAN), and network layers. This approach is not sufficiently reliable, as activation distributions may have different distributions across various datasets, models, and layers, requiring different fitting distribution to accurately capture these characteristics.
3.	In Figures 1 and 2, the activation ratios of the base model appear within a stable range, while those of the trained model fluctuate with the edges. This inconsistent visualization hinders an intuitive comparison of the activation distributions between the base and trained models within the same figure.
4. The presentation of tables should be improved and contain at least two lines.

**Questions:**

1.	Figure 3 applies the gamma distribution to approximate the negative logarithm of activation magnitudes across distinct layers and datasets, such as layer 3 of GraphTransformer on ZINC, layer 0 of GraphiT on ZINC, and layer 1 of SAN on OGBN-PROTEINS. Is this single gamma distribution capable of accurately fitting the activation distributions across these diverse scenarios?
2.	While the paper demonstrates the existence of MAs in older models like GraphiT, GraphTransformer, and SAN, is this phenomenon also observed in more recent advancements, such as SGFormer?

---

> ### Author Response · Authors · 2024-11-22
>
> The authors thank Reviewer 5kwJ for all the interesting questions.
>
> ### Weakness #1
> As shown in previous studies [[1]](https://arxiv.org/abs/2402.17762), we experiment with augmenting self-attention with additional bias terms with the idea to model such attention biases explicitly, introducing additional learnable parameters for each head:
>
> $$
> b_e = Q k e' \tag{1}
> $$
> $$
> b_v = \text{softmax}(A_e) v' \tag{2}
> $$
>
> where $k, e, v ∈ ℝ^d$ are the key, edge, and node bias terms (one per each attention head), $A_e$ is the edge attention output, and $d$ the corresponding hidden dimension. Equations (1) and (2) were chosen based on the architectures of the models considered and following validation experiments.
>
> ### Weakness #2
> Our motivation was to mathematically highlight MAs as an anomalous behavior that deviated from a standard one. Therefore, we defined a priori what the standard behavior was, i.e., the distribution of activations for an untrained base model with Xavier randomly initialized weights (Figure 3a in the main paper), on which the best approximation was precisely given by a gamma family distribution. Consequently, the same distribution had to be reused in all the other datasets/models/layers precisely to highlight anomalous behaviors such as MAs.
>
> We would like to specify that the method by which we chose the best base model distribution relies on an empirical approach, where we tried various families of distributions (e.g., normal, lognormal, exponential, Weibull). Experiments showed that the gamma family was the best approximation. The fit is always done independently for each layer, model, and dataset.
>
> ### Weakness #3
> Initially, we plotted figures displaying lines for each layer of the base and trained models. However, the authors found that displaying 20 lines in each of the nine figures was visually confusing (mostly with ZINC). To improve clarity, the layer values for the base model were transformed into a min-max range, which helped highlight the MAs in the layers of the trained model. Nevertheless, one of the original figures is included in the `supplementary_material/ICLR25_Rebuttal.pdf` as Figure 2.
>
> ### Weakness #4
> We appreciated the suggestion and we will modify tables and illustrations to meet the standards.
>
> ### Question #1
> Our motivation was to mathematically highlight MAs as an anomalous behavior that deviated from a standard one. Therefore, we defined a priori what the standard behavior was, i.e., the distribution of activations for an untrained base model with Xavier randomly initialized weights (Fig.3a on the main paper), on which the best approximation was precisely given by a gamma family distribution. Consequently, the same distribution had to be reused in all the other datasets/models/layers precisely to highlight anomalous behaviors such as MAs. We would like to specify that the method by which we chose the best base model distribution relies on an empirical approach, where we tried various families of distributions (e.g., normal, lognormal, exponential, Weibull) whose experiments showed that the gamma family was the best approximation. The fit is always done independently for each layer, model, dataset.
>
> ### Question #2
> We experimented with newer models such as GraphGPS [[2]](https://arxiv.org/abs/2205.12454) and CSA [[3]](https://arxiv.org/abs/2304.10933), which showed us the presence of MAs on both ZINC (as depicted in Figure 3 in `supplementary_material/ICLR25_Rebuttal.pdf`) and PCQM4Q-full v2 datasets.
> In addition, we will enrich the benchmark by introducing new models and datasets to further show the presence of MAs. For example, we will investigate [[2]](https://arxiv.org/abs/2205.12454) and [[3]](https://arxiv.org/abs/2304.10933) over:
> - `ogbg-molhiv`
> - `ogbg-molpcba`
> - `ogbg-ppa`
> - `ogbg-code2`
>
> The choice of models and datasets, in any way, always reflects our original idea: that is, models that incorporate extra information about edges from their datasets, via edge features, and thus focus attention on both nodes and edges during their message passing.
>
> ### References
> [1] Mingjie Sun, Xinlei Chen, J Zico Kolter, and Zhuang Liu. Massive activa-
> tions in large language models. arXiv preprint arXiv:2402.17762, 2024.
>
> [2] Ladislav Ramp´aˇsek, Michael Galkin, Vijay Prakash Dwivedi, Anh Tuan Luu,
> Guy Wolf, and Dominique Beaini. Recipe for a general, powerful, scalable
> graph transformer. Advances in Neural Information Processing Systems,
> 35:14501–14515, 2022.
>
> [3] Romain Menegaux, Emmanuel Jehanno, Margot Selosse, and Julien Mairal.
> Self-attention in colors: Another take on encoding graph structure in trans-
> formers. arXiv preprint arXiv:2304.10933, 2023

---

> > ### Author Response · Authors · 2024-11-26
> >
> > Thank you for taking the time to provide your feedback. We wanted to kindly check if you have had the opportunity to review our rebuttal and if there are any remaining concerns that we may not have addressed.

---

> > > ### Author Response · Authors · 2024-11-28
> > >
> > > Thank you for taking the time to provide your feedback. As we approach the deadline, we wanted to check if you have had the opportunity to review our rebuttal and if there are any remaining concerns we may not have addressed.

---

### Meta-Review · Area_Chair_ztqv · 2024-12-20

**Metareview:**

The authors examine Massive Activations (MAs) in graph transformers, and they propose an Explicit Bias Term (EBT) to counteract the emergence of MAs.

The reviewers recognized the importance of studying MAs within the graph learning domain, found the results insightful, and regarded the proposed EBT as an effective method.

However, they raised several significant concerns:
- The novelty and technical contribution over a similar study in the natural language domain remain unclear.
- The study focuses on graph transformers for graphs with edge features but does not address widely-used attention-based GNNs, limiting its broader relevance.
- More extensive experiments, especially with a wider range of datasets and models, are necessary to strengthen the findings.
- The theoretical analysis lacks sufficient depth.

Considering both the reviews and rebuttals, the meta-reviewer recommends rejecting the paper but encourages the authors to improve the work based on the provided feedback.

**Additional Comments On Reviewer Discussion:**

Reviewer participation during the discussion phase was limited; however, the meta-reviewer carefully read both the rebuttals and the initial reviews to make a decision.

---

### Decision · Program_Chairs · 2025-01-22

Reject